# Sweetpotato-based infant foods produce porridge with lower viscosity and aflatoxin level than cereal-based complementary blends

**Francis Kweku Amagloh** [ID]*

Department of Food Science and Technology, Faculty of Agricultural and Consumer Sciences, University for Development Studies, Tamale, Ghana

* fkamagloh@uds.edu.gh

## Abstract

The viscosity, protein, and total aflatoxins contents in orange-fleshed sweetpotato (OFSP) and cereal-based commercial complementary formulations and the effect of dilution on the protein content of the formulations were investigated. Standard procedures were used for the determination of these parameters. Over 80% of the formulations had a viscosity above the recommended consistency of 1000–3000 cP for feeding young children. The consistency of OFSP-legume porridge was significantly (2392.5 cP; $p < 0.001$) lower, about 1.7 and 3.4 times than cereal-only and cereal-legume blends, respectively. All the complementary flours, except the cereal-only, met the proposed protein requirement of 6 to 11 g per 100 g for feeding children aged 6 to 23 months on an as-is basis. However, the protein content in the porridges on an as-would-be-eaten basis was about 6% lower than the as-is basis value. About 38% of the complementary foods had total aflatoxin level above the acceptable limit of 10 ppb, mainly in blends containing peanuts, maize, or both. Adding more water to meet the required thickness of cereal-only and cereal-legume porridges diluted the protein content. More efforts are needed from regulatory bodies and all stakeholders to ensure complementary foods are safe in terms of mycotoxin levels, particularly those containing maize, peanut, or both as ingredients.

## 1. Introduction

Malnutrition remains the leading cause of child mortality in developing countries. For example, according to the World Health Organization, sub-Saharan Africa still has the highest global mortality rate of children under the age of 5; treatment and preventable efforts to curb this includes adequate nutrition and access to safe foods [1].

Complementary feeding in low-income countries in sub-Saharan Africa depends mainly on cereals [2, 3]. Due to the high dietary bulk, complementary foods require excess dilution with water during preparation to achieve a suitable consistency [4]. Excessive addition of water leads to a reduction in energy and nutrient density [5]. Thus, children are fed porridges

**Data Availability Statement:** All relevant data are within the manuscript.

**Funding:** The author received no specific funding for this work.

**Competing interests:** The author has declared that no competing interests exist.

with low nutrient- or energy-density [6, 7]. The low energy and nutrient density of complementary foods are critical contributors to childhood malnutrition and growth faltering [8]. This can be prevented through optimal complementary feeding (i.e., timely, adequate, appropriate, and safe) [8].

Complementary foods augment breast milk and should ideally have the consistency of yoghurt in a tub, be energy- and nutrient-dense, and be free of microbial contamination (live organisms, toxins, or by-products). Walker [9] proposed that porridge viscosity is a significant factor in protein-energy malnutrition. A porridge viscosity between 1000–3000 cP has been reported to be of appropriate consistency for the first solid foods for infants [7]. However, such viscosity required for complementary foods is difficult to achieve in cereals if they are fermented, roasted or blended with legumes unless germinated [10–12]. Ideally, cereals used as feed ingredients should be pre-digested by α-amylase to partially digest the starch and reduce the viscosity of the porridge [13].

Lutter & Dewey [14] proposed a protein content of 6 to 11 g per 100 g in complementary food flours for the diet of children aged 6 to 23 months. It should be noted that the nutrient content of the flour for porridge may not be the same as the porridge served. The nutrient level could be diluted by the quantity of water added to obtain a suitable consistency.

Complementary foods are also expected to be microbially safe, either by being free from microbial contamination (live organisms, toxins, or by-products) or having a contamination level below nationally and internationally acceptable limits. However, from a two-week, cross-sectional survey in most regions of Ghana, where 48 commercial complementary samples were sourced and analysed for total aflatoxins, about a third of the samples exceeded the acceptable limit of 20 ppb [15]. Thus, there should be a greater interest in finding alternative non-cereal-based complementary foods. In addition, it has been reported that cereal-legume blends, mainly made from maize and peanuts, usually have higher total aflatoxin loads [15, 16]. When fed to infants and young children, these toxin-infested foods expose them to aflatoxin and its related adverse effects. This is evidenced in a longitudinal study conducted in Benin, West Africa, among children aged 16 to 37 months, where a strong association was found between aflatoxin exposure and impaired growth, with possible effects on immunity and susceptibility to infections [17]. For this reason, countries such as Switzerland and the Netherlands have set 0.18 ppb and 0.21–0.39 ppb, respectively, as the maximum allowable limit for aflatoxin (B1) in foods for infants and young children, with the European Union value of 0.10 ppb being stricter [18]. To curb the negative effect of aflatoxin, its ingestion from food should be kept as low as possible [19]. If possible, food for infants and young children should be kept under 1.0 ppb aflatoxin contamination [18].

There is no standard for non-cereal-based complementary foods, only cereal-based [20], implying that cereals should be the base ingredient for complementary foods. This may have contributed to the over-dependency on cereals, even though other crops can be used for complementary foods, particularly in low-income countries where complementary foods for the resource-poor are prepared from locally available ingredients and are not enriched with micronutrients [21]. Therefore, there is a need to utilise other climatic-smart crops. For example, the ability of sweetpotato to be cultivated in most ecologies, being drought-tolerant once established with a relatively short maturity period of three (3) to five (5) months, makes it a climatic-smart crop [22]. In addition, an earlier study has shown that sweetpotato requires less use of energy for cooking and the addition of water than cereals in preparing complementary food recipes at the household level [23].

Complementary foods have been formulated using sweetpotato, including orange-fleshed sweetpotato (OFSP), as a base ingredient [24–27]. The reported advantages of these formulations over household-level cereal-based blends included a higher concentration of provitamin

A (plant dietary source of vitamin A), simple sugars (imparting natural sweetness), and forming less viscous porridge (less reduction of nutrient and energy density) [28]. The levels of simple sugars, especially sucrose, increase during root storage [29–31]. Additionally, maltose is formed from starch when the roots are cooked at a temperature of 65 ˚C and above [29, 31–33]. The conversion of starch to these simple sugars during storage and cooking imparts a natural sweetness to the porridge. Also, it contributes to the lower viscosity of sweet potato formulations for infants compared with cereal-based types [5].

In a review of the efficacy and effectiveness of nutrition interventions, provitamin A from sweetpotato roots was found to have better bioavailability than that from other leafy vegetables [34]. Also, OFSP complementary foods meet WHO recommendations for daily consumption of fruits and vegetables for older infants and young children [35].

This study aimed to assess the viscosity, protein and aflatoxin contents (on an as-would-be-eaten basis) in sweetpotato- and cereal-based commercial formulations sourced from Ghana and Uganda. It is hypothesised that the sweetpotato-based formulations would have lower viscosity, less protein dilution and lower total aflatoxins than cereal-based blends.

## 2. Materials and methods

### 2.1 Study design & sampling

A completely randomised design was employed in this study. Different brands ($n$ = 23) of commercial complementary foods were obtained from open markets and supermarkets in Ghana and Uganda. In addition, the OFSP-based formulations were obtained from a product exhibition at a workshop organised by International Potato Centre in 2019 in Kampala, Uganda. One (1) household-level maize-only dough was prepared from orange maize for this work. To prepare the orange maize dough, 500 g of whole orange maize grain was steeped for 74 h, wet-milled and sieved using a 630 μm mesh size. A commercial maize-based complementary food in Ghana was used as the control. The ingredients of the control formulation included whole-corn flour, skimmed milk, sucrose, palm olein, acidity regulator, calcium carbonate, vitamins, ferrous fumarate, vanillin, zinc sulphate, Bifidus culture and potassium iodide. The cereal used in the control formulation was pre-digested, denoted as CHE (cereal hydrolysed enzymatically). For ethical reasons, none of the proprietary products is referred to by brand name in this manuscript; instead, identification is based on the ingredients listed on the package.

### 2.2 Laboratory analysis

The moisture, crude protein and total aflatoxin content of the complementary flour and the orange maize dough samples were determined. The viscosity and protein levels of the prepared porridges were also measured.

**2.2.1 Moisture determination.** The analytical procedure for moisture (AOAC 925.10) published by AOAC International was slightly modified; drying was done at 108˚C overnight using a Thermo Scientific Heratherm oven [36]. The moisture determination was to correct the weights for the viscosity measurement and express the total content of aflatoxin on a dry matter basis.

**2.2.2 Crude protein determination.** The total nitrogen content was determined using the AOAC 960.52 [36], and crude protein was calculated using 6.25 as the conversion factor.

**2.2.3 Total aflatoxins determination.** Total aflatoxins in all the complementary flours were quantified using a tablet-assisted aflatoxin mobile assay (mReader) that uses Reveal Q + test strips (Neogen Corporation) as previously published [15].

**Table 1. Moisture and weights taken equivalent to 100 g dry matter content.**

| Constituent of complementary food (number of blends) | Moisture as-is (g/100 g) | Weight equivalent to 100 g dry matter content (g) | Total volume of water used to attain suitable viscosity (ml) |
|---|---|---|---|
| Maize+Maltodextrin+Skimmed milk powder+Micronutrients | 3.67 [e-g] (2.39, 4.95) | 103.81 [c,d] (101.64, 105.99) | 300.00 [g,h] (280.60, 319.40) |
| Maize+Millet+Soy | 3.02 [f,g] (1.75, 4.31) | 103.14 [c,d] (100.96, 105.31) | 335.67 [e-h] (316.27, 355.06) |
| Maize+Millet+Soy+Micronutrients | 3.98 [d-g] (2.70, 5.26) | 104.14 [c,d] (101.97, 106.32) | 403.00 [b,c] (383.60, 422.40) |
| Maize+Millet+Soy+Peanut | 4.23 [c-g] (2.95, 5.51) | 104.42 [b-d] (102.25, 106.60) | 387.30 [b-d] (367.90, 406.70) |
| Maize+Millet+Wheat+Rice+Soy+Peanut | 4.88 [d-g] (3.60, 6.16) | 105.14 [b-d] (102.96, 107.31) | 337.33 [d-h] (317.94, 356.73) |
| Maize+Soy[a] | 3.98 [d-g] (3.08, 4.89) | 104.16 [c,d] (102.62, 105.70) | 409.20 [b] (395.50, 422.90) |
| Maize+Soy+Amaranth+Beetroot | 7.70 [b,c] (6.42, 8.98) | 108.35 [b,c] (106.17, 110.52) | 347.00 [d-g] (327.60, 366.40) |
| Maize+Soy+Peanut[b] | 4.34 [d-g] (3.61, 5.08) | 104.56 [b-d] (103.30, 105.82) | 364.42 [c-e] (353.22, 375.62) |
| Millet+Soy[a] | 6.85 [b-d] (5.95, 7.76) | 107.39 [b,c] (105.85, 108.93) | 350.02 [d-f] (336.30, 363.73) |
| OFSP+Maize+Pea | 9.01 [b] (7.73, 10.29) | 109.91 [b] (107.73, 112.08) | 300.00 [g,h] (280.60, 319.40) |
| OFSP+Rice+Soy | 7.29 [b-d] (6.01, 8.57) | 107.87 [b,c] (105.69, 110.04) | 310.40 [f-h] (291.00, 329.80) |
| OFSP+Sorghum+Soy | 6.45 [b-e] (5.17, 7.73) | 106.89 [b-d] (104.72, 109.07) | 345.67 [d-g] (326.27, 365.06) |
| OFSP+Soy+Yellow maize | 6.72 [b-e] (5.44, 8.00) | 107.21 [b-d] (105.03, 109.38) | 300.00 [g,h] (280.60, 319.40) |
| Fermented orange maize | 48.97 [a] (47.69, 50.25) | 196.18 [a] (194.01, 198.36) | 482.00 [a] (462.60, 501.40) |
| Rice+Soy | 7.66 [b,c] (6.38, 8.94) | 108.31 [b,c] (106.14, 110.49) | 361.60 [c-e] (342.20, 381.00) |
| Rice+Wheat+Soy | 4.88 [c-g] (3.60, 6.16) | 105.14 [b-d] (102.96, 107.31) | 383.83 [b-e] (364.44, 403.23) |
| Wheat+Maltodextrin+Skimmed milk powder+Micronutrients[a] | 2.82 [g] (1.91, 3.72) | 102.90 [d] (101.37, 104.44) | 300.00 [g,h] (286.30, 313.70) |
| Wheat+Soy | 6.08 [b-f] (4.80, 7.36) | 106.47 [b-d] (104.30, 108.65) | 371.67 [b-e] (352.27, 391.06) |
| Yellow maize+Wheat+Millet+Soy+Coconut | 5.02 [c-g] (3.74, 6.30) | 105.28 [b-d] (103.11, 107.46) | 369.33 [b-e] (349.94, 388.73) |
| P-value | <0.0001 | <0.0001 | <0.0001 |

Values are means of triplicate measurement per number of sample (95% confidence interval for the mean). Means that do not share a letter are significantly different.

[a]Wheat + Maltodextrin + Skimmed milk powder+Micronutrients; Millet + Soy; and Maize + Soy: Two (2) different brands.

[b]Maize + Soy + Peanut: Three (3) different brands.

All others were one (1) brand.

**2.2.4 Viscosity measurement.** The weight of each flour or dough equivalent to 100 g dry matter was determined (Table 1) and used to prepare the porridge according to the manufacturer's instructions or as it is conventionally done for the orange-maize dough in Ghana. For ease of handling, apart from the product with maize, maltodextrin, skimmed milk powder, micronutrients or CHE printed on the package, the weight used for all the others was equivalent to 50 g dry matter. To minimise the effect of temperature on viscosity measurement using a DVE digital viscometer (Ametek Brookfield, Korea) at 100 rpm, the prepared porridges were carefully transferred to a 600-mL glass beaker in a water bath with intermittent stirring to avoid the formation of air bubbles at the bottom until the temperature was 40°C. The initial viscosity was measured with an average torque of 47% and varying spindle numbers (2 to 5) depending on the consistency of the porridge.

After the first viscosity measurement, the porridges were diluted with water to a viscosity of 1000 to 3000 cP, which corresponded to the range of consistency of the control samples. Spindle number 2 at a rotational speed of 100 rpm and an average torque of 62% was used during dilution to adjust the viscosity of the porridges to the suitable consistency. The volume of water used to achieve the right consistency of the porridges required for complementary feeding was measured by the difference between the initial and final volumes of water used for the dilution. Triplicate readings were taken for each aliquot sampled for the porridge preparation.

## 2.3 Statistical analysis

One-way analysis of the variance procedure in Minitab®16.2.2 (Minitab Inc., State College, PA, USA) was used to compare the means for each parameter assayed. In addition, Tukey's studentised range test was employed when the mean comparison test was significant (p < 0.05) to ascertain which treatment means differed.

# 3. Results and discussion

## 3.1 Viscosity of prepared porridge and after dilution

Nearly 80% of the samples had viscosities exceeding the drinking consistency of 1000–3000 cP before dilution (Fig 1A). The viscosity of the sweetpotato-legume porridge was significantly (p < 0.001; 2392.5 cP) lower than that of cereal-only and cereal-legume porridges. The consistency in sweetpotato-legume porridge was respectively about 1.7 and 3.4 times lower than cereal-only and cereal-legume porridges but 1.1 times higher than the control. The viscosity of the OFSP-based porridges obtained in this study was in the range reported by Araro et al. [37]. The viscosity trend for the ready-to-consume cereal-based porridges confirms previous work on complementary foods [38].

The lower viscosity values recorded for the OFSP-legume blends were in line with those reported by Jemberu et al. [39] for similar OFSP-legume mixtures. In their study, they observed a reduction in viscosity values of formulated porridges as the proportion of OFSP flour was increased. This observation was likely due to the high content of simple sugars in the sweetpotato flour [5, 39].

According to Copeland et al. [40], high viscosity is attributed to the characteristic swelling of starch during cooking, which causes it to gelatinise. This limits the food intake of the child. Therefore, it is desirable to reduce the viscosity of ready-to-consume porridge. This is achieved by adding a certain amount of diluent (usually water), which results in reduced energy and protein density.

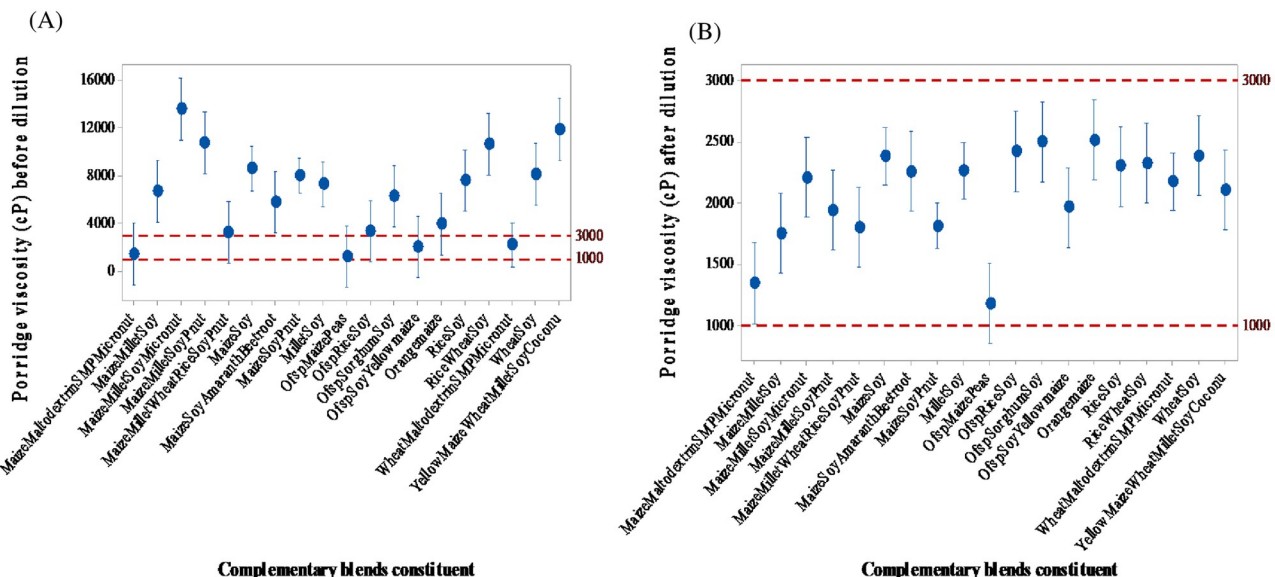

**Fig 1. (A) Viscosity of commercial complementary blends before dilution and (B) the viscosity of commercial complementary blends after dilution.**

The cereal-based porridges with similar viscosity as the OFSP-based samples contained cereal in the form of CHE. The α-amylase hydrolyses starch to dextrin and maltose during the pre-digestion, reducing the consistency of the thick cereal porridges and enhancing their energy and nutrient density [41]. Interestingly, the OFSP-based products produced by small-scale manufacturers that did not use CHE as an ingredient had comparable viscosity to CHE-containing products. The relatively low viscosity of OFSP-based porridges confirms previous reports by Ridley et al. [31] and Nabubuya et al. [42]. The endogenous β-amylase in OFSP led to self-viscosity thinning. The thinner the porridge at preparation, the more "food" will be added to obtain a suitable porridge viscosity, which invariably increases nutrient density.

The viscosity values of the porridges after dilution (Fig 1B) showed no significant difference (p = 0.273) since the goal was to achieve the desired consistency of the control sample. It could be argued that amylolytic flour (produced from germinated cereals) produces a less viscous porridge when made at the household level. Previous researchers have shown that it is not effective when added before cooking the slurry for porridge, as the enzyme will be denatured by heat. Although it can be added to porridge, microbial safety concerns have been highlighted [7]. Furthermore, the porridge prepared from fermented orange-maize has a higher tendency for the starch to associate and retrograde, forming a thick paste upon cooling. This is not desirable as such products require additional water to obtain a viscosity suitable for feeding, which further reduces the energy and nutrient density.

## 3.2 Protein content of complementary flours and dough on as-received and as-would-be-eaten bases

The protein content (g/100 g) on an as-is basis of fermented orange-maize dough was significantly lower (3.85; 95% CI = 1.19, 6.52) than that of the other complementary flours investigated (Fig 2). Apart from the cereal-only (fermented orange-maize) flour, all other

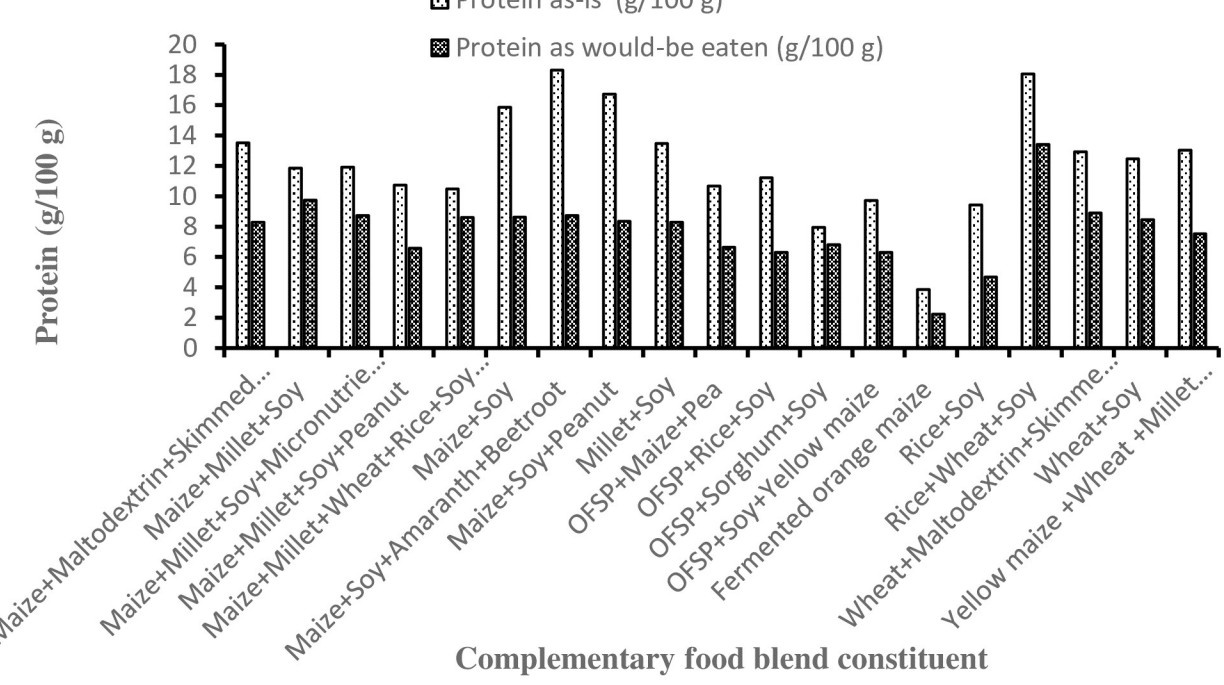

**Fig 2. The protein content of commercial complementary blends.**

complementary flours met the suggested protein content of 6 to 11 g for feeding children 6 to 23 months of age [14]. As expected, protein content in the porridges on as-would-be-eaten basis was almost 1.6-times lower than on as-received basis (7.9 *vs*. 12.8; p < 0.001). The protein content of orange-maize porridge is similar to that observed by Alamu et al. [43], who also reported lower (1–3.7%) protein content of traditional maize-based complementary foods in Zambia.

Therefore, future research efforts should focus on how best to meet the energy and nutrient requirements, but also take into account factors that affect energy intake, such as the energy density and viscosity of complementary foods.

## 3.3 Volume of water used for dilution to get appropriate consistency for complementary feeding

All but the control sample and those containing maltodextrins were wet-cooked into porridge before serving. The quantity of water used to obtain the appropriate consistency of 1000–3000 cP [7] for the same amount of complementary food is presented in Table 1. The amount of water needed to dilute the complementary blends to obtain the proper drinking consistency varied significantly (p < 0.0001) among the mixtures, with the cereal-only porridge recording the largest volume (482 ml). The water required to dilute the OFSP-based complementary blends to a drinking consistency ranged from 300–345 ml, which was lower than the cereal-based complementary blends. The volume of water added to attain the suitable viscosity is crucial because more water means lower energy and nutrient density, which is the case with most cereal-based complementary foods in Africa. This finding supports Amagloh & Andrade [23], who reported that less water was required in the preparation of the OFSP-based porridge than for Weanimix, a maize-based complementary food in Ghana.

Therefore, OFSP could be regarded as a climate-smart food ingredient for complementary foods because it cooks quickly [44] and requires less water during cooking [23].

## 3.4 Total aflatoxins contamination

The level of aflatoxin contamination in the complementary foods sampled is shown in Fig 3. Generally, about 38% of the foods had aflatoxin levels above 10 ppb, while 21% contained aflatoxins above 20 ppb. All samples with contamination above the U.S. tolerable limit of 20 ppb were cereal-legume mixtures containing either peanuts, maize or both, as reported in a previous study by van Egmond et al. [45]. Using the African median permissible limit of 10 ppb for processed foods [45], as in the case of many international jurisdictions, 63% of the commercial baby foods examined in this study are still not considered wholesome even for adult consumption. It is particularly worrying that 25% of the proprietary complementary foods examined in this study had contamination levels above 20 ppb.

However, these results are not surprising, as in a similar study [15], about 60% of 48 complementary foods in the Ghanaian market had aflatoxin contamination above 20 ppb. Unfortunately, parents/caregivers may purchase these foods on the market, often with limited resources to support children's growth and overall health, only to inadvertently expose them to aflatoxins that could be detrimental to their health. These high levels of aflatoxin contamination in complementary foods underscore the importance of enhanced regulatory measures. It is reasonable to assume that either the raw materials used for these foods are highly contaminated with the toxin, or the toxin multiplies in the product due to favourable storage conditions [46]. It is also noteworthy that the complementary foods sampled were all above more stringent maximum allowable limits for aflatoxin (B1) in infant foods in Switzerland (0.18 ppb), the Netherlands (0.21–0.39 ppb) and the European Union (0.10 ppb) [18]. This is a

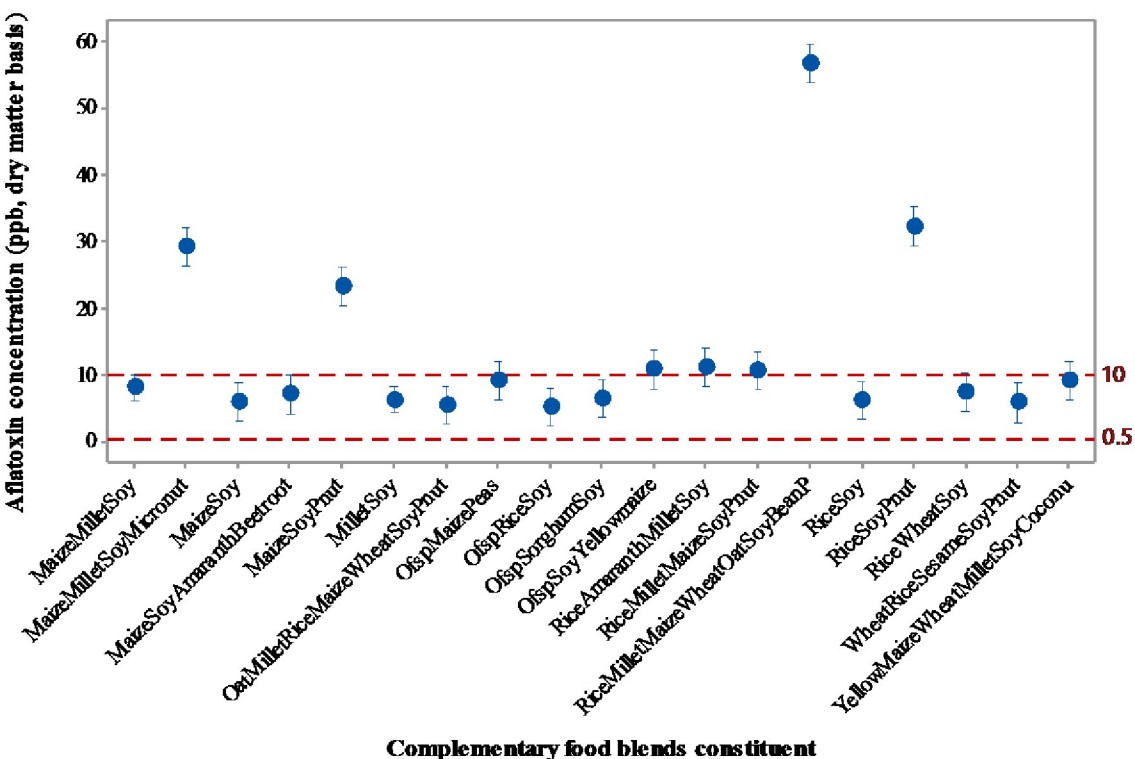

**Fig 3. Aflatoxin concentration of commercial complementary blends.**

grave public health concern that needs urgent attention from regulatory bodies in the sub-region.

The effect of dilution on other nutritional properties (apart from protein content), though expected to be reduced during the dilution, was not included in this study due to financial constraints and is recognised as a study limitation.

## 4. Conclusions

The OFSP-containing blends made porridge with appropriate viscosity that did not require further dilution with water as was required for most of the cereal-legume combinations. Apart from the cereal-only porridge, the complementary foods had sufficient protein content to meet the recommendation for 6 to 23 months children The complementary foods, with the exception of the OFSP-legume blend, had consistencies that greatly exceeded the recommended range for young children. Adding more water to produce the required viscosity will likely reduce energy and nutrient density significantly. Furthermore, the high aflatoxin contamination in the complementary food blends containing maize, peanuts or both underscores the importance of enhanced regulatory measures to address this public health concern. Based on the present findings, OFSP could be a better alternative ingredient for complementary foods, particularly for resource-poor households, as it requires less dilution with water, has an appropriate viscosity when prepared as a porridge and is associated with lower total aflatoxin content. However, more efforts are needed from regulatory bodies and all stakeholders to ensure that complementary foods, especially those containing maize, peanuts, or both as ingredients, meet local and international standards for aflatoxin contamination.

## Acknowledgments

The author will like to acknowledge Ms. Lilian Abban and Ms. Zeinab Mahamuda for their efforts during the data collection of the study.

## Author Contributions

**Conceptualization:** Francis Kweku Amagloh.

**Data curation:** Francis Kweku Amagloh.

**Formal analysis:** Francis Kweku Amagloh.

**Investigation:** Francis Kweku Amagloh.

**Methodology:** Francis Kweku Amagloh.

**Resources:** Francis Kweku Amagloh.

**Software:** Francis Kweku Amagloh.

**Validation:** Francis Kweku Amagloh.

**Visualization:** Francis Kweku Amagloh.

**Writing – original draft:** Francis Kweku Amagloh.

**Writing – review & editing:** Francis Kweku Amagloh.

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
