## [Decision Letter · Decision Letter 0]

4 Aug 2022

PONE-D-22-17748Sweetpotato-based infant food as an alternative complementary formulationPLOS ONE

Dear Dr. Amagloh,

Thank you for submitting your manuscript to PLOS ONE. After careful consideration, we feel that it has merit but does not fully meet PLOS ONE’s publication criteria as it currently stands. Therefore, we invite you to submit a revised version of the manuscript that addresses the points raised during the review process. Please submit your revised manuscript by Sep 18 2022 11:59PM. If you will need more time than this to complete your revisions, please reply to this message or contact the journal office at plosone@plos.org. Please include the following items when submitting your revised manuscript:A rebuttal letter that responds to each point raised by the academic editor and reviewer(s). You should upload this letter as a separate file labeled 'Response to Reviewers'.A marked-up copy of your manuscript that highlights changes made to the original version. You should upload this as a separate file labeled 'Revised Manuscript with Track Changes'.An unmarked version of your revised paper without tracked changes. You should upload this as a separate file labeled 'Manuscript'.

We look forward to receiving your revised manuscript.

Kind regards,

Fatih Oz, Ph.D.

Academic Editor

PLOS ONE

Journal Requirements:

Additional Editor Comments:

Dear Author,

Thank you for submitting your manuscript to Plos One. I have completed my evaluation of your manuscript. As appended below, the reviewers have raised major concerns/critiques (Reviewer #1 is against publication) and suggested further justification/work to consolidate the findings. Do go through the comments and amend the manuscript accordingly.

Reviewers' comments:

Reviewer's Responses to Questions

**Comments to the Author**

1. Is the manuscript technically sound, and do the data support the conclusions?

Reviewer #1: Partly

Reviewer #2: Yes

Reviewer #3: Partly

2. Has the statistical analysis been performed appropriately and rigorously? 

Reviewer #1: Yes

Reviewer #2: Yes

Reviewer #3: Yes

3. Have the authors made all data underlying the findings in their manuscript fully available?

Reviewer #1: No

Reviewer #2: Yes

Reviewer #3: No

4. Is the manuscript presented in an intelligible fashion and written in standard English?

Reviewer #1: No

Reviewer #2: Yes

Reviewer #3: No

5. Review Comments to the Author

Reviewer #1: Although the aims of the study are good, it was planned and carried out very superficially. It has been seen that the written language, sample, analyzes and methods of construction are not sufficient. Samples are not well explained, not understandable. It is thought that measuring each of the samples in the viscosity study by bringing them to the same dry matter significantly affects the difference between the samples. It is thought that the evaluation made in this way will not give correct results.

Reviewer #2: PONE-D-22-17748-peer-review-v1.comments

Sweetpotato-based infant food as an alternative complementary formulation

Comment 1: The research is quite innovative and interesting, with clearly established objectives and conclusions.

Comment 2: Abstract: The following line is little bit confusing, it is recommended to rephrase it to make it more clear and easily understandable: ‘However, the protein content in the porridges on an as-would-be-eaten basis was about 6% lower.’

Comment 3: Abstract: Have you mentioned the amount and the desirable water content for proper mixing of the blends?

Comment 4: 62: Is not 10 ppb really high even to be marked as maximum level for infant cereals, when we are talking about the acceptable levels as 0.18-0.39 ppb?

Comment 5: Line No. 109: Did you measures moisture and protein content only in nutritional parameters? If so, what about the determination of other nutritional contents like minerals, fiber and vitamins?

Comment 6: Line No. 150: Have you mentioned anywhere about total number of samples or treatments that you have used?

Comment 7: Line No. 223: In my opinion this level of alfatoxin as reported from the results here are high for infant consumption. Please add justification.

Comment 8: Line No. Conclusion: Overall the study is good, but in this type of study a small animal based efficacy investigation should also be included as to ensure pros and cons of the product developed. Without having its safety confirmation we are unable to recommend such innovative idea.

Reviewer #3: The research article “Sweetpotato-based infant food as an alternative complementary formulation” by Francis seeks to describe the combination of different ingredients and their properties and product studies (protein and viscosity).

The article needs extensive corrections.

I think the author should change the title it did not match with the content.

Comment 1: First, the English language has to be corrected, the authors use too much every-day spoken English, too much for a scientific publication.

Comment 2: Authors should add the descriptive sensory analysis or 9-point hedonic scale and shelf life of porridge.

Comment 3: The author should add the physico-chemical properties of multigrain porridge such as fat, crude fiber, ash, and carbohydrates

Comment 4: Author should check the dough's rheological properties which are prepared from orange maize.

Comment 3: The abbreviation CHE have no sense in line number 104-105, the abbreviation details should be added in line number 101-104 (The ingredients of the control formulation included whole corn flour (CHE indicated on the package), skimmed milk, sucrose, palm olein, acidity regulator, calcium carbonate, vitamins, ferrous fumarate, vanillin, zinc sulphate, Bifidus culture, and potassium iodide.)

Comment 6: In line number 114 the author should clearly add which type of dry process and instrument were used for drying at 108°C overnight.

Comment 7: Rewrite the viscosity measurement protocol from line number 124-142.

Comment 8: In Table 1 no need for the repetition of (n=1), you can add it at end of the table

Example: Data are presented as mean ± SD (n = 3)

a-nMeans with the same superscript in a column do not vary significantly (p<0.05) from each other

In table 1 author adds the value please justify then and explain in paragraph also. Or the author removes those values.

Comment 9: The author checks the result discussion and conclusion part.

6. PLOS authors have the option to publish the peer review history of their article (what does this mean?). If published, this will include your full peer review and any attached files.

Reviewer #1: No

Reviewer #2: No

Reviewer #3: No

---

## [Author Response · Author response to Decision Letter 0]

10 Aug 2022

Response to queries in italics: Reference to revised manuscript with track changes

Reviewer reports:

Reviewer #1: 

Although the aims of the study are good, it was planned and carried out very superficially. It has been seen that the written language, sample, analyzes and methods of construction are not sufficient. Samples are not well explained, not understandable. It is thought that measuring each of the samples in the viscosity study by bringing them to the same dry matter significantly affects the difference between the samples. It is thought that the evaluation made in this way will not give correct results.

Your comments are very much appreciated. However, the samples were existing proprietary products (lines 107 – 108); thus, their brand names were undisclosed for ethical reasons. Therefore, the best way to describe these samples without disclosing brand names was to indicate their constituent ingredients, as captured in lines 107 – 108. Regarding analyses and methods, some revisions have been made (lines 103, 105 – 106, 131 – 135). 

Although what you have suggested is right for getting the actual viscosities of the samples. It is required, in rheological analyses such as visco-analysis, that samples are corrected to the same dry matter to avoid wrong comparisons as a result of varied initial moisture contents. With the same dry matter content, the viscosity of the samples can best be compared, hence the reason to correct the samples to the same dry matter content in this study.

Reviewer #2: PONE-D-22-17748-peer-review-v1.comments

Comment 1: The research is quite innovative and interesting, with clearly established objectives and conclusions.

Thank you very much for your kind remarks.

Comment 2: Abstract: The following line is little bit confusing, it is recommended to rephrase it to make it more clear and easily understandable: 'However, the protein content in the porridges on an as-would-be-eaten basis was about 6% lower.'

Thank you for your comment. The statement has been rephrased: ' However, the protein content in the porridges on an as-would-be-eaten basis was about 6% lower than the as-is basis value.'

Comment 3: Abstract: Have you mentioned the amount and the desirable water content for proper mixing of the blends?

No, please, reason being the word limit for the abstract. However, it has been stated in the materials and methods section (lines 139 – 142).

Comment 4: 62: Is not 10 ppb really high even to be marked as maximum level for infant cereals, when we are talking about the acceptable levels as 0.18-0.39 ppb?

Yes, you are very right. The statement has been revised as "To curb the negative effect of aflatoxin, its ingestion from food should be kept as low as possible (19)." 

Comment 5: Line No. 109: Did you measures moisture and protein content only in nutritional parameters? If so, what about the determination of other nutritional contents like minerals, fiber and vitamins?

Yes, please. Moisture and protein contents were the only parameters measured. The study did not include other nutritional properties due to financial constraints, which was captured as a limitation (lines 248 – 250). 

Comment 6: Line No. 150: Have you mentioned anywhere about total number of samples or treatments that you have used?

Yes, please. The total number of samples (24) was mentioned in line 97. 

Comment 7: Line No. 223: In my opinion this level of alfatoxin as reported from the results here are high for infant consumption. Please add justification.

You are right. The aflatoxin level reported is for infant consumption. Lines 243 – 247 underscores the importance that the complementary blends assessed were not safe for infant consumption as all of them had toxin level above the permissible limit for infants as per Switzerland, Netherlands, and European standards and the need for stricter regulatory measures. 

Comment 8: Line No. Conclusion: Overall the study is good, but in this type of study a small animal based efficacy investigation should also be included as to ensure pros and cons of the product developed. Without having its safety confirmation we are unable to recommend such innovative idea.

Your comment is highly appreciated. However, the products used for this study were on the market, and were not developed by the author. Thus, the study could not include any animal-based investigations.

Reviewer #3: 

The research article "Sweetpotato-based infant food as an alternative complementary formulation" by Francis seeks to describe the combination of different ingredients and their properties and product studies (protein and viscosity).

The article needs extensive corrections.

Your comment is greatly appreciated. Some revisions have been made to that effect.

I think the author should change the title it did not match with the content.

Thank you very much. The title has been changed as suggested to "Sweetpotato-based infant foods have lower viscous porridge and total aflatoxins level than cereal-based complementary blends" in lines 1 – 2. 

Comment 1: First, the English language has to be corrected, the authors use too much every-day spoken English, too much for a scientific publication.

Thank you for your comment. The safety and quality of complementary foods are critical; thus, there is the need to present it in such a way that the general populace can read and understand the manuscript. I am also finding it difficult where the "every-day spoken English" phrases are in the manuscript.

Comment 2: Authors should add the descriptive sensory analysis or 9-point hedonic scale and shelf life of porridge.

The author appreciates your comment. However, the samples used in this study were existing products on the market and not developed by the author. Hence, sensory analysis and shelf-life studies were not relevant. These are done during the product development phase.

Comment 3: The author should add the physico-chemical properties of multigrain porridge such as fat, crude fiber, ash, and carbohydrates.

Thank you. The author would have loved to determine the aforementioned physico-chemical properties. However, the study did not include these properties due to financial constraints, as indicated in lines 248 – 250. Moreover, being proprietary products, the nutritional information was on the package. The cereal-only sample, particularly from white-coloured maize, has been reported in previous work to be nutritionally inappropriate for complementary feeding.

Amagloh, F. K., Weber, J. L., Brough, L., Hardacre, A., Mutukumira, A. N., & Coad, J. (2012). Complementary food blends and malnutrition among infants in Ghana–A review and a proposed solution. Scientific Research and Essays, 7(9), 972-988. doi:10.5897/SRE11.1362 

Comment 4: Author should check the dough's rheological properties which are prepared from orange maize.

The viscosity measured for the (fermented) orange maize is one of the rheological properties.

Comment 3: The abbreviation CHE have no sense in line number 104-105, the abbreviation details should be added in line number 101-104 (The ingredients of the control formulation included whole corn flour (CHE indicated on the package), skimmed milk, sucrose, palm olein, acidity regulator, calcium carbonate, vitamins, ferrous fumarate, vanillin, zinc sulphate, Bifidus culture, and potassium iodide.)

Comment well noted and the revision has been done appropriately. Lines 103, 105 & 106.

Comment 6: In line number 114 the author should clearly add which type of dry process and instrument were used for drying at 108°C overnight.

Thank you. The instrument used was a Thermo Scientific Heratherm oven. This has now been added to the revised manuscript (lines 115 – 116) as suggested.

Comment 7: Rewrite the viscosity measurement protocol from line number 124-142.

Thank you. Please, some revisions have been made as suggested, as captured in lines 131 – 135, 138 – 139.

Comment 8: In Table 1 no need for the repetition of (n=1), you can add it at end of the table

Example: Data are presented as mean ± SD (n = 3)

a-nMeans with the same superscript in a column do not vary significantly (p<0.05) from each other

In table 1 author adds the value please justify then and explain in paragraph also. Or the author removes those values.

Your comments are well noted. Please, revisions have been made to that effect. Lines 223-225.

Comment 9: The author checks the result discussion and conclusion part.

Thank you very much. The author has re-checked the abovementioned areas as suggested.

---

## [Decision Letter · Decision Letter 1]

31 Aug 2022

PONE-D-22-17748R1Sweetpotato-based infant foods have lower viscous porridge and total aflatoxins level than cereal-based complementary blendsPLOS ONE

Dear Dr. Amagloh,

Thank you for submitting your manuscript to PLOS ONE. After careful consideration, we feel that it has merit but does not fully meet PLOS ONE’s publication criteria as it currently stands. Therefore, we invite you to submit a revised version of the manuscript that addresses the points raised during the review process.

 Please take into consideration the comments raised by Reviewer 1. The language has to be checked by a native speaker. This will be the last option for revision. 

We look forward to receiving your revised manuscript.

Kind regards,

Fatih Oz, Ph.D.

Academic Editor

PLOS ONE

Additional Editor Comments:

Dear Authors,

Please take into consideration the comments raised by Reviewer 1. The language has to be checked by a native speaker. This will be the last option for revision.

Reviewers' comments:

Reviewer's Responses to Questions

**Comments to the Author**

1. If the authors have adequately addressed your comments raised in a previous round of review and you feel that this manuscript is now acceptable for publication, you may indicate that here to bypass the “Comments to the Author” section, enter your conflict of interest statement in the “Confidential to Editor” section, and submit your "Accept" recommendation.

Reviewer #1: (No Response)

Reviewer #2: All comments have been addressed

Reviewer #3: All comments have been addressed

2. Is the manuscript technically sound, and do the data support the conclusions?

Reviewer #1: Partly

Reviewer #2: Yes

Reviewer #3: Partly

3. Has the statistical analysis been performed appropriately and rigorously? 

Reviewer #1: Yes

Reviewer #2: Yes

Reviewer #3: Yes

4. Have the authors made all data underlying the findings in their manuscript fully available?

Reviewer #1: No

Reviewer #2: Yes

Reviewer #3: Yes

5. Is the manuscript presented in an intelligible fashion and written in standard English?

Reviewer #1: No

Reviewer #2: Yes

Reviewer #3: Yes

6. Review Comments to the Author

Reviewer #1: Unfortunately, my views on the study before the revision have not changed. The study was planned and carried out very superficially. It has been seen that the written language, sample, analyzes and methods of construction are not

sufficient. Samples are not well explained, not understandable. The author made few changes in the article and could not give adequate answers to the comments and suggestions of other referees due to various reasons.

Reviewer #2: You have revised this paper in a very good way. Thanks a lot for your great work. Wish you all the best.

Reviewer #3: The author addresses all comments. But the author should take the comments in a positive way which helps their manuscript to make more effective.

7. PLOS authors have the option to publish the peer review history of their article (what does this mean?). If published, this will include your full peer review and any attached files.

Reviewer #1: No

Reviewer #2: **Yes: **Rana Muhammad Aadil

Reviewer #3: No

---

## [Author Response · Author response to Decision Letter 1]

9 Sep 2022

Response to queries in italics 

Reviewer reports:

Reviewer #1: 

Unfortunately, my views on the study before the revision have not changed. The study was planned and carried out very superficially. It has been seen that the written language, sample, analyzes and methods of construction are not sufficient. Samples are not well explained, not understandable. The author made few changes in the article and could not give adequate answers to the comments and suggestions of other referees due to various reasons.

The author appreciates your comments and has made some revisions to that effect in track changes. I have attached comments from a commercial proofreader for your perusal.

I also got Prof Jane Coad, a British, who made some comments I have used to revise the manuscripts.

I also made other comments and can be seen as track changes

However, regarding the correction of the samples to the same dry matter, attached are some published articles that also used a similar approach for viscosity determination (Boulemkahel et al., 2021; Iwe et al., 2016; Nawaz et al., 2016; Wang et al., 2013). The author hopes these revisions and references meet your expectation and kind consideration. 

The author has shared pictures of the complementary blends from which the author selected some for this study. However, this should not be made public as the author did not obtain approval or consent from the companies.

Reviewer #2: PONE-D-22-17748-peer-review-v1.comments

Comment: You have revised this paper in a very good way. Thanks a lot for your great work. Wish you all the best.

Thank you very much for your great remarks.

Reviewer #3: 

Comment: The author addresses all comments. But the author should take the comments in a positive way which helps their manuscript to make more effective. 

Thank you very much for your constructive comments. Your advice is well taken.

References

Boulemkahel, S., Betoret, E., Benatallah, L., & Rosell, C. M. (2021). Effect of low pressures homogenization on the physico-chemical and functional properties of rice flour. Food Hydrocolloids, 112, 106373.

Iwe, M. O, U. Onyeukwu, U, & A.N. Agiriga, A. N | Yildiz, F. (2016). Proximate, functional and pasting properties of FARO 44 rice, African yam bean and brown cowpea seeds composite flour, Cogent Food & Agriculture, 2:1, DOI: 10.1080/23311932.2016.1142409

Nawaz, M. A, Fukai, S., & Bhandari, B. (2016). Effect of Different Cooking Conditions on the Pasting Properties of Flours of Glutinous Rice Varieties from Lao People’s Democratic Republic, International Journal of Food Properties, 19:9, 2026-2040, DOI: 10.1080/10942912.2015.1092163

Wang L, Deng F, Ren WJ, Yang WY. (2013). Effects of shading on starch pasting characteristics of indica hybrid rice (Oryza sativa L.). PLoS One. Jul 5;8(7):e68220. doi: 10.1371/journal.pone.0068220. PMID: 23861872; PMCID: PMC3702574

---

## [Editor Report · Decision Letter 2]

20 Sep 2022

Sweetpotato-based infant foods produce porridge with lower viscosity and aflatoxin level than cereal-based complementary blends

PONE-D-22-17748R2

Dear Dr. Amagloh,

We’re pleased to inform you that your manuscript has been judged scientifically suitable for publication and will be formally accepted for publication once it meets all outstanding technical requirements.

Kind regards,

Fatih Oz, Ph.D.

Academic Editor

PLOS ONE
---

## [Editor Report · Acceptance letter]

4 Oct 2022

PONE-D-22-17748R2 

Sweetpotato-based infant foods produce porridge with lower viscosity and aflatoxin level than cereal-based complementary blends 

Dear Dr. Amagloh:

I'm pleased to inform you that your manuscript has been deemed suitable for publication in PLOS ONE. Congratulations! Your manuscript is now with our production department. 

Kind regards, 

on behalf of

Professor Fatih Oz 

Academic Editor

PLOS ONE